# Sensor-based localization of epidemic sources on human mobility networks

**Jun Li**[1☯¤a], **Juliane Manitz**[1☯¤b], **Enrico Bertuzzo**[2], **Eric D. Kolaczyk**[1]*

**1** Department of Mathematics & Statistics, Boston University, Boston, MA, United States of America,
**2** Dipartimento di Scienze Ambientali, Informatica e Statistica, University of Venice Cà Foscari, Italy

☯ These authors contributed equally to this work.
¤a Current address: Google, Mountain View, CA, United States of America
¤b Current address: EMD Serono, Billerica, MA, United States of America
* kolaczyk@bu.edu

**Data Availability Statement:** Data are available from https://github.com/jmanitz/NetOrigin-data.

**Funding:** JL and EDK were supported in part by awards from the US Air Force Office of Scientific Research (12RSL042) and Army Research Office

## Abstract

We investigate the source detection problem in epidemiology, which is one of the most important issues for control of epidemics. Mathematically, we reformulate the problem as one of identifying the relevant component in a multivariate Gaussian mixture model. Focusing on the study of cholera and diseases with similar modes of transmission, we calibrate the parameters of our mixture model using human mobility networks within a stochastic, spatially explicit epidemiological model for waterborne disease. Furthermore, we adopt a Bayesian perspective, so that prior information on source location can be incorporated (e.g., reflecting the impact of local conditions). Posterior-based inference is performed, which permits estimates in the form of either individual locations or regions. Importantly, our estimator only requires first-arrival times of the epidemic by putative observers, typically located only at a small proportion of nodes. The proposed method is demonstrated within the context of the 2000-2002 cholera outbreak in the KwaZulu-Natal province of South Africa.

## Author summary

Tracking the source of an epidemic outbreak is of crucial importance as it allows for identification of communities where control efforts should be focused for both short and long-term management and control of the disease. However, such identification is often problematic, time-consuming, and data-intensive. Recently network-based analysis approaches have been established for source detection to account for complex modern spreading, driven substantially by human mobility. Here we develop a probabilistic framework for waterborne disease, that allows investigators to infer the community or the region sparking an outbreak based on a sparse surveillance network. The framework can integrate prior information on the likelihood of a community being the source, for instance as a function of population size or hygiene conditions. Furthermore, we assign an accuracy measure to the resulting source estimate, which is crucial for its practical usability. We test the method in the context of the 2000-2002 cholera outbreak in the Kwa-Zulu-Natal province with promising results. Moreover, we outline a series of guidelines in

(W911NF-15-1-0440). JM was supported in part by the Alexander von Humboldt Foundation. The funders had no role in study design, data collection and analysis, decision to publish, or preparation of the manuscript.

**Competing interests:** The authors have declared that no competing interests exist.

terms of data needs and preliminary operations to implement the proposed framework in practice.

This is a *PLOS Computational Biology* Methods paper.

## Introduction

One of the most important factors in epidemic control is to trace the source or origin of an epidemic [1, 2]. This problem is sometimes called 'source localization' (and can in fact involve multiple sources). Ideally, one would like to locate the source based on data capturing the entire history of the epidemic, including times of infection / recovery of individuals as well as information on contact between individuals and of individuals with infective aspects of the environment (e.g., water sources). However, epidemic history is complex and high-dimensional, and almost invariably the data are incomplete—often substantially so [3, 4].

Over the past 5-10 years, researchers have found it useful to reformulate the localization problem as that of estimating a source node(s) on a complex network. There have been a large number of contributions in this area to date. A recent and comprehensive review has been conducted by [5]. Many approaches use network-distance-based measures of centrality to identify the source node in a complex network, such as rumor centrality [6, 7] or Jordan centrality [8, 9]. A related idea is that of effective distance-based source detection [10, 11]. However, typically these methods assume network-wide observation of the infection status of nodes at either a single time point or a handful of such snapshots, which is generally unrealistic for large networks—particularly in the context of human disease. Alternatively, sensor-based methods are designed to instead locate a source based on arrival-time information of infection from only a subset of observer nodes (e.g., [12–14]).

Despite the development in this area, there is still substantial room for improvement [5]. In general, methods proposed to date frequently fail to assimilate the often-abundant information that can be gained through epidemic modeling, as well as additional prior information. In addition, they typically do not provide measures of uncertainty quantification. Both of these aspects are especially important in the context of human disease, where policy providers and decision makers are often data-poor and yet required to make concrete decisions that have pronounced impact on society. In this paper, focusing on the illustrative example of cholera epidemics, we propose a method of source detection that integrates (i) a sensor-based approach, with (ii) a stochastic differential equation model for water-borne disease. In turn, we adopt a Bayesian framework, thus allowing for uncertainty quantification and the formal use of prior information.

A key component of our approach is the incorporation of human mobility networks. Human mobility is one of the main drivers for the spreading of infectious diseases. Understanding, predicting and possibly controlling the propagation of an epidemic in a population cannot prescind from the analysis of the underlying human mobility patterns. Historically, network-based research incorporating human mobility has focused on infectious diseases transmitted through direct contact between individuals e.g. [15–18]. However, the role of human mobility in the spreading of waterborne diseases (where transmission is mediated by water) has also recently attracted increasing attention e.g. [19–22]. Indeed, a susceptible individual can be exposed to contaminated water while travelling or commuting and seed the infection in the resident community once back. On the other hand, asymptomatic infected individuals (who potentially shed pathogens but whose movement is not impaired by the

disease symptoms) can spread pathogens while moving among different human communities. These two mechanisms highlight the critical role of human mobility as a notion extending beyond direct contact.

In this paper, we recast the source detection problem as one of identifying the relevant mixture component in a multivariate Gaussian mixture model from [12]. Human mobility within the stochastic, spatially-explicit epidemiological model of [22] is used to calibrate the parameters. Our estimator requires only first-arrival times of the epidemic at a small proportion of nodes, termed sensors or observers. Adopting a Bayesian perspective opens the possibility to seamlessly integrate available nontrivial prior knowledge from previously observed spreading patterns or other data sources. Moreover, we are able to quantify uncertainty in the resulting estimators. Specifically, our approach provides (a) statistically well-defined region(s) of nodes that are likely to be the spreading origin of the observed process, accompanied by a corresponding posterior probability.

We develop and apply our method in the context of the 2000–2002 cholera outbreak in the KwaZulu-Natal province, South Africa. In particular, our integrative, Bayesian approach demonstrates significant improvement in this context over the use of a generic sensor-based source detection approach alone [12].

To better place our contributions in context, we note the following points in comparison to related work in the literature. First, while there are a number of network-based methods of epidemic source detection that are not generic and that incorporate some knowledge of disease epidemiology (e.g., [23–25]), this is the first article to integrate cholera-specific transmission models in network-based source detection. Second, while human mobility networks have been used previously in network-based epidemic source detection (e.g., [26], who also use a gravity model similar to ours), this is the first article to integrate the role of human mobility in the complex spreading of waterborne diseases. Finally, while a number of Bayesian approaches have been suggested or developed in epidemic source detection (e.g., [23, 24, 27]), to the best of our knowledge none of these have developed an informed prior probability distribution. In addition, while our use of generic networks for the underlying spreading pattern is less common in the literature (in contrast to assuming a tree-like structure), there is indeed precedent (e.g., [14, 25]).

Code implementing our proposed method has been integrated into the *NetOrigin* package in R.

## Results

### Sensor-based source localization: Overview of proposed method

We assume a network $G = (V, E)$ to be given that is composed of a set of nodes $v \in V$ that are inter-connected by links $(u, v) \in E$. Furthermore, there is a spreading process on this network, which originates in source node $s^* \in V$. For pre-defined sensors at a small fraction of nodes, $O = \{o_k\}_{k=1}^K, K \leq |V|$, we observe the first-arrival times of the spreading process, i.e. $\mathbf{t} = (t_1, \ldots, t_K)^\top$. In the epidemiological context motivating our work, the set of nodes $v \in V$ are human communities, and the first-arrival times are the time points at which a given level of disease incidence is attained in observed communities. Our aim is to develop a good estimator for the source $s^*$ and to quantify the uncertainty in that estimator.

Conditional on the underlying spreading process and a given source $s^*$, the first-arrival times $\mathbf{t}$ are assumed to follow a $K$-dimensional multivariate Gaussian distribution. The *a priori* chance that a given node $s$ is the epidemic source is modeled according to a prior distribution $\boldsymbol{\pi} = (\pi_1, \ldots, \pi_N)^\top$ over network nodes $v \in V$ with $\sum_{v=1}^N \pi_v = 1$, where $N = |V|$ is the total number of nodes. Through this prior we incorporate subjective beliefs or other sources of information about the origin of the spreading process. Statistical inference on source location is then

based on the corresponding posterior, with the most probable source determined as

$$\hat{s} = \arg \max_{s \in V} P(S = s | \boldsymbol{t}), \tag{1}$$

and the most probable region, by

$$\hat{C} = \{s : P(s | \boldsymbol{T} = t) > \tau_\alpha\}, \text{ so that } \sum_{s \in \hat{C}} P(s | \boldsymbol{T} = t) \geq 1 - \alpha, \tag{2}$$

where here $\alpha \in (0, 1)$ is pre-specified and $\tau_\alpha \in (0, 1)$ is the largest such threshold for which the conditions in (2) hold.

The underlying spreading process is modeled using a set of stochastic differential equations for the spread of water-borne disease, consisting of three main elements. First, fundamentally, our model is a version of the well-known susceptible-infected-removed (SIR) model, but expanded to differentiate rates of death cross each class of individuals as well as to include components for both symptomatic and asymptomatic infection. Most of the rate parameters are simple constants to be set by the user (e.g., using historical data, public records, etc.). However, second, the rates of (a)symptomatic infection are modeled proportional to a 'force of infection' term which, for a given location, summarizes the aggregate contribution of bacterial concentration at neighboring locations and the extent of human mobility from the latter to the former location. Finally, third, the bacterial concentration at each location is modeled using a linear differential equation that includes a term reflecting the number of infected individuals and the volume of the local water reservoir.

Human mobility is represented through the network $G$, which is taken to be directed and weighted. Here nodes correspond to communities and weights on links between nodes reflect the probability of movement by individuals from one node to another. In our applications, these probabilities are calculated using a simple gravity model, combining information on the size of communities and the distance between them.

Our overall approach to sensor-based source localization combines a Bayesian extension of the method in [12] with the human mobility portion of the spreading model in [20]. By construction, the source localization problem in our setting effectively reduces to that of identifying through the posterior distribution the relevant component in a multivariate Gaussian mixture model. The necessary parameters for the individual Gaussian components, i.e., the means and covariances of the first-arrival times observed at sensors 1, . . ., $K$, are calibrated using a combination of stochastic simulation from our spreading model and statistical smoothing of the corresponding output. These simulations in turn are run using various rate parameters whose values are retrieved through literature review. Additional details regarding modeling and implementation can be found in Methods.

## Analysis of the 2000–2002 South African cholera outbreak

We applied our source estimation approach to data from the 2000–2002 cholera outbreak in the KwaZulu-Natal province, South Africa. The outbreak lasted for two years, starting in August of 2000, and ultimately involved about 140, 000 recorded cases in two major waves in the respective summers [28]. Fig 1 shows the epidemic curve. We can see the peak of the first wave is much higher than the peak of the second wave.

Figs 2 and 3 show a spatial representation of some of the data and results relevant to our model. These data have already been described in detail in [20]. In the figures are shown the spatial locations of $N = 851$ communities in the KwaZulu-Natal province, indicated by dots for which the area scales with population size. In turn, each community corresponds to a node in our human mobility network $G$. A visualization of this network is also provided, as an overlay.

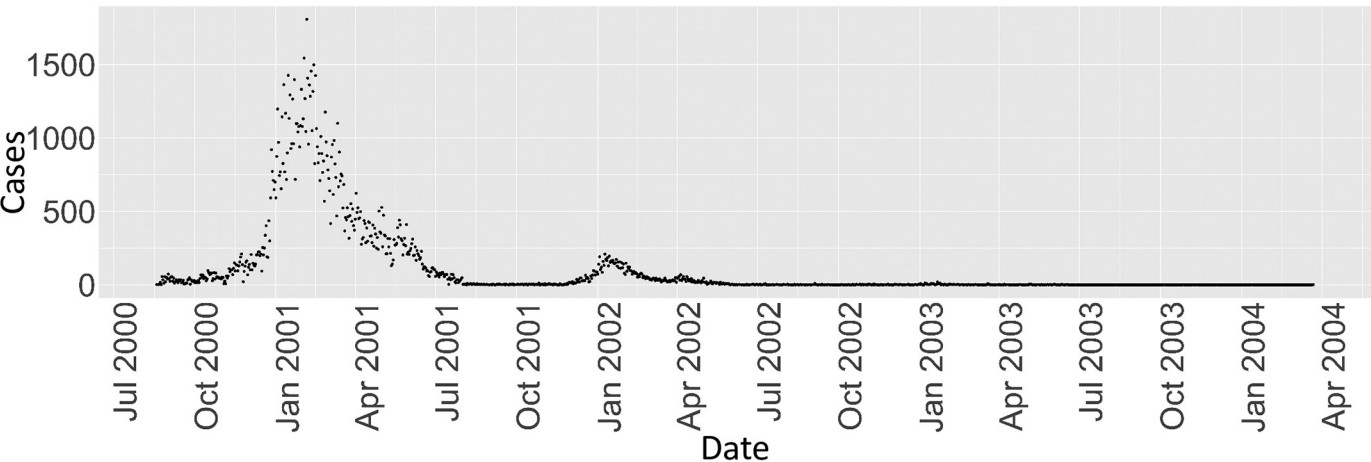

**Fig 1. Number of cases per day in the 2000-2002 KwaZulu-Natal cholera epidemic.**

In order to improve interpretability, only the three most frequented outbound links are shown (typically corresponding to roughly 10% of the outward mobility from a node). Among those links, we split them into 2 sets: the links with top 10% weights and those with bottom 90% weights. For the first set, we kept their weights unchanged; for the second set, we decreased their weights to 10% of their original value. That is sufficient to illustrate several characteristics of the network. In particular, we note the local, grid-like connectivity of much of the network, which is then complemented by a handful of nodes with substantially higher and more global connectivity. The network visualization suggests small-world behavior, which can be confirmed through computational methods applied to the underlying human mobility network (see S1 File). The more highly connected nodes with global connectivity correspond roughly to (i) Durban, the largest city in KwaZulu-Natal, and other cities in the Greater Durban Municipality (e.g., Inanda); (ii) Pietermaritzburg, the capital and second-largest city in KwaZulu-Natal, situated 80 km inland from Durban; and (iii) Newcastle, the third largest city, located near the northwest edge of the province.

Also represented in Figs 2 and 3 is a local version of the basic reproduction number, $R_0$, for each community, through appropriate shading of the nodes. In epidemiology, the basic reproduction number of an infection can be thought of as the number of cases to derive from one infected case on average over the course of its infectious period, in an otherwise uninfected population [2]. In a well-mixed population, when $R_0 < 1$, the infection will die out in the long run. On the contrary, if $R_0 > 1$, the infection will spread. In the case of multiple interconnected local populations, the concept of basic reproduction number has been generalized by [29, 30]. Here, a local version of the reproduction number has been computed for each node, following the approach of [20], which combines information on community size with models for contamination and exposure rates that incorporate access to (in)adequate toilet facilities and to water, respectively. See Methods for additional details.

For each wave, nine nodes with highest weighted degree (also called node strength) in the human mobility network were chosen, from among those nodes that were infected during a given wave, to serve the role of 'observers' (or sensor nodes) in our source detection algorithm. These represent roughly 1% of the total nodes in the underlying network for each wave. Selecting observer nodes based on degree is expected to improve detection accuracy. (We examine this assertion further in the synthetic experiments described later in this section.) We see that the two resulting sets of observer nodes are largely complementary in nature. The observer

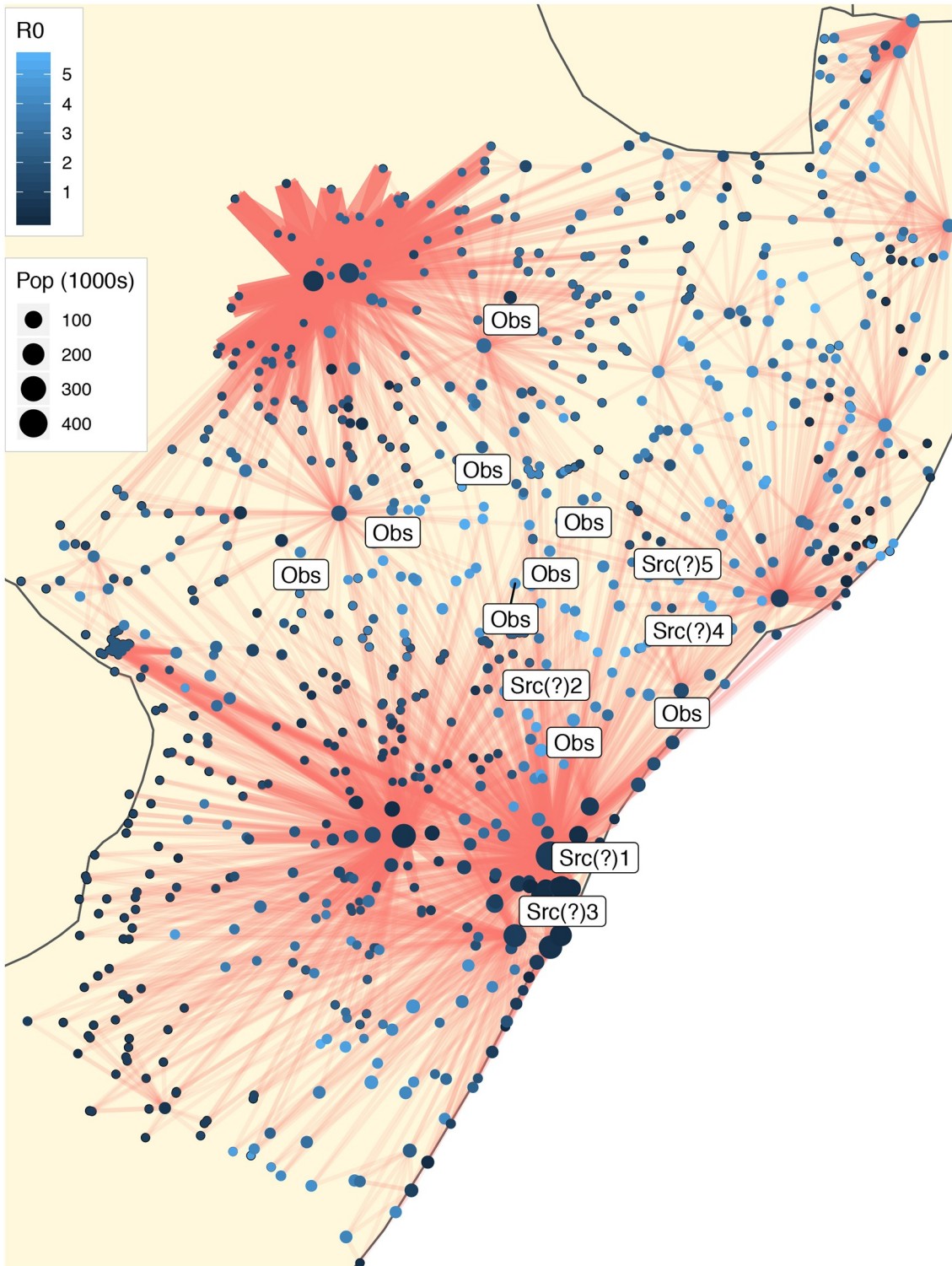

**Fig 2. The human communities of the KwaZulu-Natal province and their corresponding mobility network (showing only three most frequently outbound links), with nodes sized and colored to indicate population and $R_0$, respectively.** Labels indicate nine 'observer' nodes and top five putative sources for Wave 1 of the cholera epidemic, as identified by our proposed methodology (based on a uniform prior).

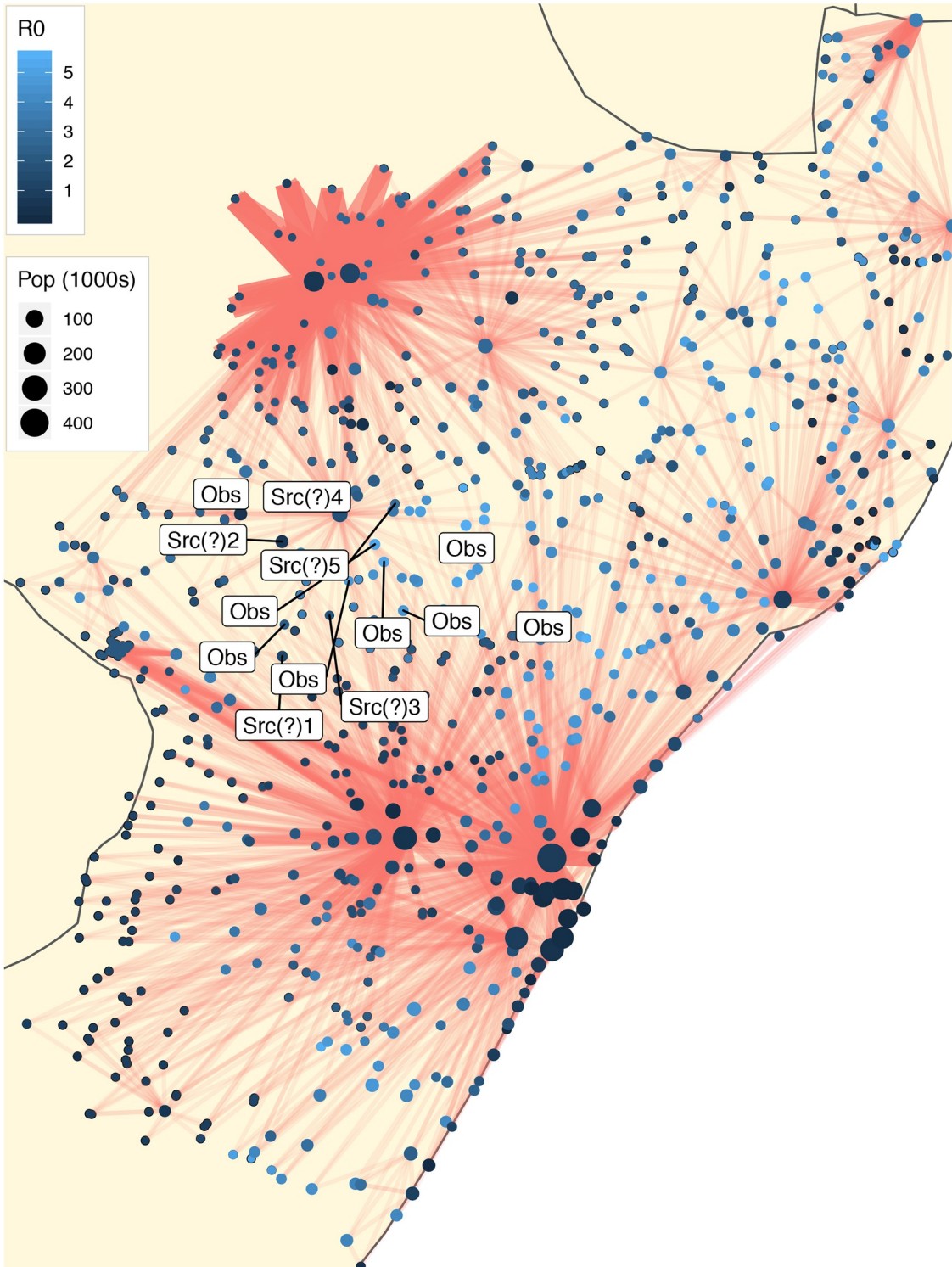

**Fig 3. The same human communities mobility network as in Fig 2, but with observers and putative sources corresponding to analysis of Wave 2 of the cholera epidemic.** (Note: The fourth putative source is also an observer node).

nodes for Wave 1, shown in Fig 2, are spread throughout much of the province, to the north and northwest of Durban / Pietermaritzburg and to the south and southeast of Newcastle. In contrast, the observer nodes for Wave 2, shown in Fig 3, are concentrated almost entirely between these two major metropolitan regions.

Now consider the results of our source detection methodology applied to these data. Shown in Fig 4 are the posterior probabilities for those ten nodes found to have the largest chance of being a source node, for each of Wave 1 and 2, under both a uniform prior on nodes and a prior proportional to the local $R_0$. For each of the four combinations of wave and prior, the corresponding ten nodes ended up representing a most probable posterior region of roughly 0.70 posterior mass. In comparing results for the two waves, there is clear evidence that the posterior in Wave 1 is substantially more concentrated on just one ($R_0$ prior) or two (uniform prior) nodes. On the other hand, while in Wave 2 there is some evidence of similar concentration under the uniform prior, with the $R_0$ prior there is comparatively less information in the posterior to differentiate the ten most probable nodes. Accordingly, we see that incorporating prior information in the form of the local reproduction numbers (which in turn reflect a combination of community size with contamination and exposure rates) has a substantive impact on the shape of the corresponding posterior distributions.

To understand the impact of these differences in posterior shape on the rankings of putative sources (and, hence, the potential impact on decisions of policy, resources, etc.), consider the plots in Fig 5. Overall, it would seem that the ranks are fairly stable, with seven and eight of those nodes ranked top-10 under the uniform prior still remaining in the top-10 under the $R_0$ prior, for Waves 1 and 2, respectively. However, there are important exceptions. For example, in Wave 1, there are four points whose initial rankings change considerably by including $R_0$ information (three of which drop well out of the top-10), all of which have very small $R_0$ (i.e., 0.21 or less). Interestingly, one of these four corresponds to the top-ranked node under the uniform prior, which nevertheless remains top-10 under the $R_0$ prior (i.e., ranked 7th), suggesting that the evidence in the data towards it being a source is particularly strong. On the other hand, another of these nodes drops from third to $16^{th}$.

Although there is no ground truth for these data, some conjectures can be made based on these results. As can be seen from the map in Fig 2, these two nodes (i.e., the first and third ranked putative sources under the uniform prior) are in fact geographically quite close together and located in the vicinity of Durban. They correspond roughly to Town Verulam and Westville, respectively. The first is located on the coast about 27 kilometers north of Durban, and the second, about 10 kilometers to the west of Durban. Both have comparatively large populations and low $R_0$. In contrast, the node corresponding to an area called Eshane has a small population ($\sim 500$) with a very large $R_0$ (7.18), and yet is ranked the second most likely source under either choice of prior. Eshane is about 45 kilometers east of Greytown, a town situated on the banks of a tributary of the Umvoti River in a fertile area that produces timber, and which sits at the nexus of multiple regional routes (i.e., R33, R74 and R622) which might help the waterborne disease, cholera, to spread. Examination of the epidemic time course for Wave 1 shows that the wave was first found to spread largely along the coast (see S1 Fig). Together, therefore, these observations suggest that the results of our analysis of Wave 1 can be interpreted as saying that either (i) the epidemic originated in the interior (near Eshane) and was brought to the coast, or (ii) it in fact originated on the coast (just outside of Durban).

In comparison, the results of our analysis for Wave 2 seem to tell a consistent story, whether under the uniform prior or the $R_0$ prior, in that the two most likely putative sources are the same for either choice of prior (albeit with their order switched) and are located fairly close together. As seen from the map in Fig 3, application of our methodology indicates that the epidemic source for this wave lies inland, in the more sparsely populated central region of the

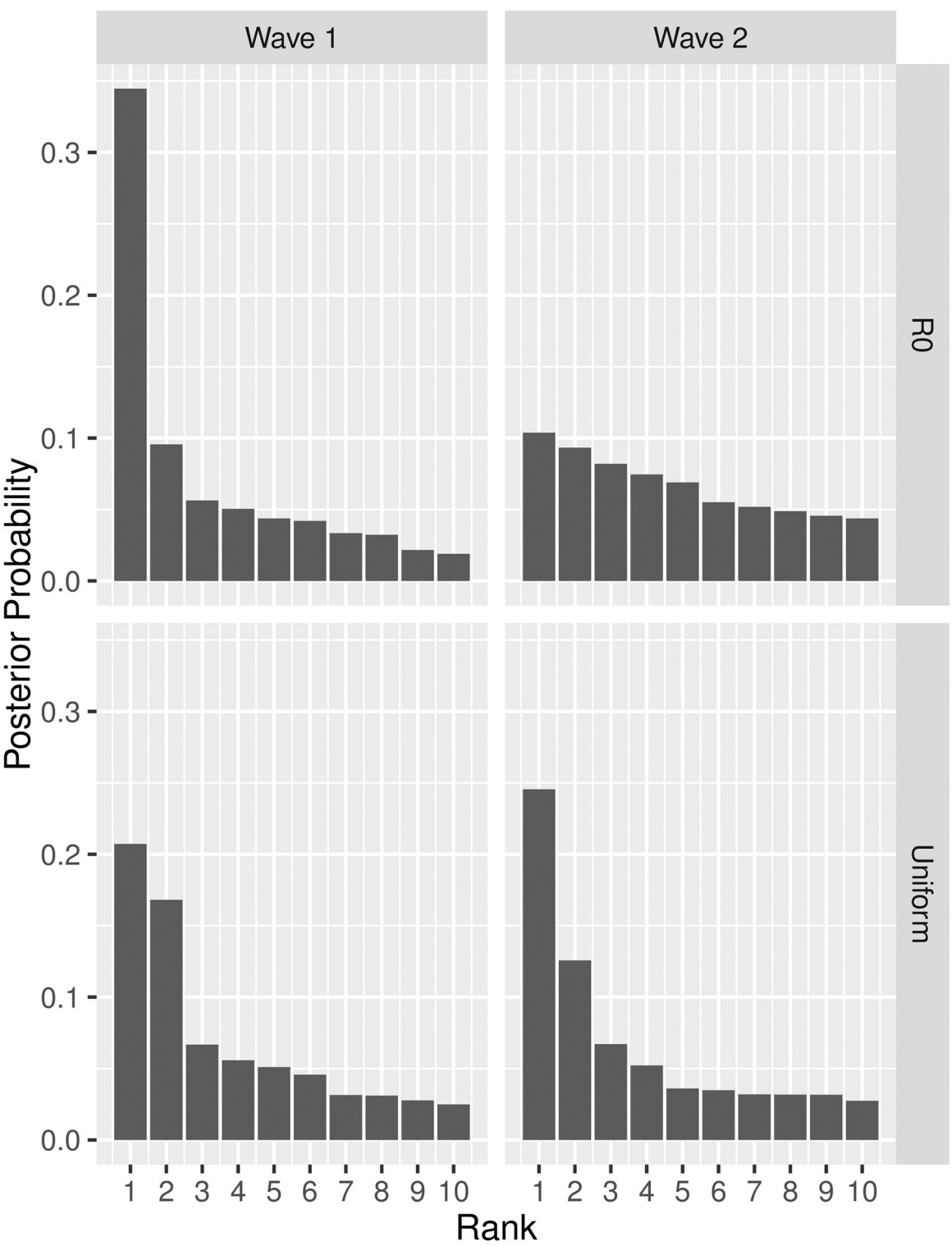

**Fig 4. Ten largest posterior probabilities** $P(S = s|t)$ **(sorted by magnitude) for source detection during Waves 1 (left) and 2 (right) of the 2000–2002 cholera epidemic in KwaZulu-Natal, South Africa, under a uniform prior (bottom) and a prior proportional to local** $R_0$ **(top).**

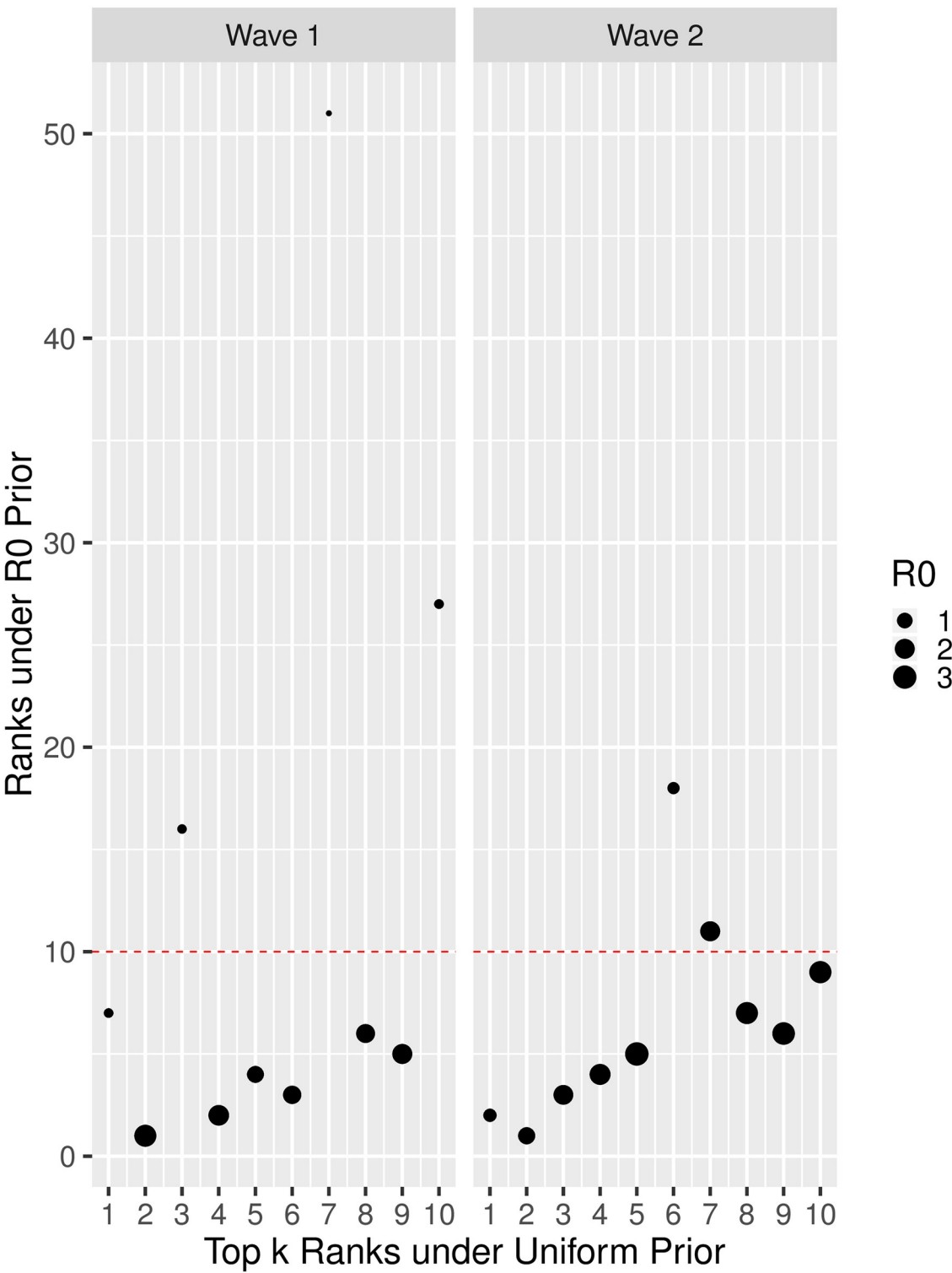

**Fig 5. Visualization of the extent to which posterior-based ranking of the top ten putative source nodes, under the uniform prior (x-axis), change when using the $R_0$ prior instead (y-axis), for Waves 1 (left) and 2 (right).** The size of each point in the scatterplots is in proportion to the $R_0$ value for the corresponding node.

province between the two major metropolitan areas of Durban / Pietermaritzburg and New-castle. Under the uniform prior, the most likely source is Ezakheni E, a town of moderate population size and somewhat elevated $R_0$ (i.e., 1.67). And about half as likely is the town of Estcourt, about 60 kilometers south, also of moderate (although smaller) population size but with substantially larger $R_0$ (i.e., 3.62). Alternatively, under the $R_0$ prior, these two nodes have nearly equal (and lower) posterior probability of being a source. Examination of the epidemic time course for Wave 2 shows that, while the earliest reported cases were to the north and northeast of the region surrounding these two nodes (i.e., near Newcastle), the bulk of the infections during this wave seemed to concentrate in this region (see S2 Fig). Therefore, the results of our analysis suggest that this second wave most likely originated from this central region, between the two larger metropolitan areas, and spread outward from there, perhaps through the system of rivers flowing through the area (i.e., Ezakheni E and Estcourt lie close to the rivers Kliprivier and Boesmansrivier, respectively).

In the above analyses, a node was said to be infected once the prevalence (i.e. the number of infected individuals) first exceeds 0.1% of the population. Additional analysis shows that our results (i.e., the top-ten ranked nodes) remain robust when this choice of threshold is decreased to 0.09% and even 0.05%, but deteriorates at 0.01% (see S3 Fig). At the same time, thresholds larger than 0.1%, even 0.2%, makes the inference procedure fail, since not all observers are infected. Formally, this failure could be avoided through appropriate adjust-ments to the underlying formulas and procedures (i.e., accounting for right-censoring in the data for these nodes), but one would still nevertheless expect a deterioration in performance.

## Synthetic experiments

In order to gain some insight into the reliability of the above results, we conducted a simula-tion study in the context of the 2000–2002 cholera outbreak. Specifically, we used the genera-tive model underlying our methodology (described above and in Methods) to generate a collection of synthetic outbreaks in order to

- investigate the impact on source estimation performance of changes in certain fundamental implementation details; and

- compare the proposed method with a comparable established approach [12] (see Discussion).

We simulated $N = 851$ scenarios, where each node was allowed to be the epidemic source in turn. A given source node was infected at Day 1 and the first arrival-time of the epidemic at an arbitrary node is defined as the day on which the prevalence (i.e. the number of infected indi-viduals) first exceeds 0.1% of the population. For each scenario, we then generated 400 realiza-tions, 300 of which were used for training (i.e. estimating the spreading parameters and in turn calibrating the model) and the remaining 100 of which were used for testing, allowing us to compare the accuracy with which source estimates matched the true underlying sources.

We investigated the robustness in performance of our methodology, as a function of chang-ing the number of observers, using different observer placement strategies, and incorporating prior knowledge or not. Specifically, we varied the

1. number of observers: 9 or 18 observers representing 1% or 2% of the total number of nodes, respectively.

2. observer placement strategies: random observer selection (random) or high-degree observer placement strategy (high-degree) [12], i.e. selecting observers with the highest (weighted) node degrees in the human mobility network.

3. incorporation of prior knowledge: informative prior, where the prior is proportional to each node's $R_0$ ($R_0$ prior), and non-informative prior, following a uniform distribution (uniform). Nodes with larger $R_0$ are easier to be infected, so it is reasonable to let the prior be proportional to these values.

Accuracy of our methodology was quantified using the following four criteria with respect to the true source $s^*$:

1. the probability that the 0.95 credible region contains $s^*$;

2. the size of the 0.95 credible region;

3. the probability that $s^*$ is ranked among the Top 10;

4. the mean distance between $s^*$ and the estimated source $\hat{s}$.

We classified the nodes into different groups according to the magnitudes of their $R_0$ (divided into 6 categories $[0, 0.9)$, $[0.9, 1.8)$, $[1.8, 2.7)$, $[2.7, 3.6)$, $[3.6, 4.5)$, $[4.5, \infty)$) and their populations (on a log base 10 scale, divided into three categories), and made box plots for the above four criteria as a function of the various conditions. See S4, S5 and S6 Figs.

Based on our simulation results, we can conclude the following:

1. The high-degree observer placement strategy outperforms the random placement strategy.

2. The frequency with which the true source $s^*$ is ranked in the Top 10 increases, and the mean distance between the true source and the estimation decreases, with increasing number of observers. At the same time, there is also a small decrease in the coverage probability of the 95% credible region and a much larger decrease in the size of the 95% credible region.

3. When using the high-degree placement of observers, the performance corresponding to 9 observers and that corresponding to 18 observers are comparable.

4. Use of a prior proportional to $R_0$ yields better results than a uniform prior when the source has large $R_0$.

5. Using either prior (uniform prior or prior proportional to $R_0$), empirical coverage probabilities of the 0.95 credibility regions are good ($> 0.7$) for sources with not too small $R_0$'s ($> 1.8$) or moderate population ($\log_{10} > 3.5$).

6. It is possible for the 0.95 confidence sets to contain over 100 nodes. However, these sets will be substantially smaller (e.g., 10's of nodes) and have good coverage probabilities if the $R_0$ of the sources are not too small ($> 1.8$) and the population is large ($\log_{10} \geq 4.5$).

7. For the probability of the true source being in the Top 10 to be at least 0.5, under a uniform prior, the $R_0$ of sources should not be too small ($> 1.8$) and the population should be large ($\log 10$-base $\geq 4.5$). Under a prior proportional to $R_0$, the $R_0$ of sources should be larger (over 2.7).

8. Using either prior (uniform or proportional to $R_0$), if the true source has moderate $R_0$ ($\geq 2.7$) and large population ($\log_{10} \geq 4.5$), the distance between true and estimated sources can be smaller than 50 (km).

In general, as can be expected, our proposed method has good performance when the source node/city has moderate or large $R_0$ and population. These simulation results also suggest the following guidelines for usage of our methodology in practice:

1. To monitor spreads of epidemics, placing resources onto transportation hubs, i.e. 'high-degree' nodes is preferred.

2. There are 79.1% nodes having $R_0 > 1.8$ or moderate population ($\log_{10} > 3.5$). Thus although we will not know the $R_0$ and the population of the true source beforehand, there are still large chances we can ensure good ($> 0.7$) coverage probabilities.

3. There are only about 5% nodes with large population ($\log_{10} \geq 4.5$) thus the chance that we have small credible region (10's of nodes) with reasonable coverage is small. However, if we use 18 observers, in most cases we can ensure good coverage (as Item 5 in the conclusion list describes) with credible regions less than 100 nodes, which are usually feasible.

4. If we use a prior proportional to $R_0$, there is a large chance that the probability of the true source being in the Top 10 to be at least 0.5, which is 41.5% (We only need the source to have large($> 2.7$) $R_0$, compared to using the uniform prior—where there is only less than 5% chance that the source fulfills the requirements described in Item 7 in the conclusion list.

## Discussion

Tracking the source of an epidemic outbreak is of crucial importance in epidemiology. Indeed, the identification of the area or the human community that sparked an outbreak is useful not only for the short-term disease control, i.e. focusing interventions in the area in an effort to stop the transmission, but also for the long-term management of the disease as such an area could be the designated target of future interventions to curb the risk of new outbreaks. Therefore, the source detection problem is relevant not only in real-time, but also retrospectively on past data. However, the correct identification is often impaired by the lack of widespread and efficient surveillance networks, especially in developing countries. Even in the cases where such health infrastructures exist, the simple analysis of the data to identify the area where the first cases where reported might lead to an incorrect identification of the true source. In fact, the real beginning of an outbreak could go unreported because initial cases are misdiagnosed. This is the case for instance with cholera, for which lab confirmation of suspected cases is typically performed routinely only when an ongoing outbreak is declared. In this context, thus, mathematical models for source identification are of primary importance.

In this paper, we developed a framework that allows the probabilistic identification of the source based on first-arrival times of the infection on a small subset of nodes (e.g. human communities) used as observers, thus potentially reducing the cost to set up and maintain a surveillance network. From a mathematical perspective, we recast the source detection problem as identifying a relevant mixture component in a multivariate Gaussian mixture model. The framework is complemented by a stochastic spatially-explicit epidemiological model that embeds information about the human mobility network and is used to calibrate the parameters characterizing the probability distributions of first arrival-times. With our approach we address the major challenges stated by [5]. Building on the sensor-based Gaussian mixture approach, our data needs are realistic for practical settings. Additionally, the implementation is computationally feasible in large networks. Furthermore, we allow generic networks for the underlying spreading pattern (in contrast to assuming a tree-like structure). Moreover, adopting a Bayesian perspective opens the possibility to seamlessly integrate available nontrivial prior knowledge from previously observed spreading pattern or other data sources. While there are many methods for source detection, it is comparatively more rare that they also quantify the estimator accuracy, and none with an informed Bayesian prior probability

distribution. Also note that our uncertainty quantification does not take into account uncertainty in the model components including gravity model and human mobility, epidemiological model for cholera spread, etc. These components are almost certainly idealized and only, at best, useful rather than correct. Thus practitioners should interpret the resulting numbers as guiding decision making rather than absolute truths. We define (a) statistically well-defined region(s) of nodes that are likely to be the spreading origin of the observed process. Because this region need not be contiguous, it also arguably provides some information on the prospect for multiple sources (although we do not formally solve here the problem of detecting multiple sources, which is notably more complex).

Among existing methods in the literature, our method can perhaps be viewed as closest to the seminal work of Pinto *et al.* [12], which has been shown to be quite competitive with many other methods under a variety of scenarios [5]. However, for the specific context of water-borne diseases studied here, our method substantially outperforms that of Pinto *et al.* in simulation (see S7 Fig). This advantage illustrates the value-added yielded by our use of highly informative prior information, i.e., through (i) utilization of the full human mobility network, (ii) encoding of prior information on quality of water and toilet facilities, and (iii) integration of a stochastic spatially-explicit epidemiological model to calibrate the means and covariances in our Gaussian mixtures.

The capability of the proposed method is demonstrated in the context of the 2000-2002 cholera outbreak in the KwaZulu-Natal province, through analyses of both actual data from the outbreak and a corresponding collection of synthetic experiments. In the experiments, we showed that the proposed method performs well if the source has moderate or larger $R_0$ and population. Examination of experimental output suggests that the decay in performance in the case of small $R_0$ or population may be due to a lack of fit with the assumed multivariate Gaussian in the mixture model at the core of our framework. While simulation suggests that the Gaussian can be quite reasonable otherwise (see S8 and S9 Figs), the use of more general mixture models may help (e.g., nonparametric Bayesian mixtures [31]). However, it is not immediately apparent how best to integrate such models with an underlying epidemiological model. Alternatively, one might instead specify a mixture of epidemiological models, each defined conditional on a different node being the source. However, posterior-based inference of the source under this approach is likely to be nontrivial to implement, since even just parameter estimation in a single such version of our underlying epidemiological model has been found to require the use of sophisticated Markov chain Monte Carlo algorithms [20]. Accordingly, our proposed approach—detecting sources through posterior-based inference in Gaussian mixture models, with mean and covariance parameters informed by epidemiological models—may be viewed as a compromise that allows for increased interpretability and computational efficiency, arguably blending statistical and mathematical modeling in the spirit of data assimilation techniques.

We note that our analysis of the KwaZulu-Natal data is a retrospective study in nature—we effectively work from those nodes with sufficiently high prevalence and infer 'backwards' through the human mobility network to putative sources. Importantly, those nodes with insufficient prevalence do not contribute to the analysis (i.e., the difference in observer times with these nodes is right-censored and hence effectively infinity). A prospective study would potentially yield different results, depending on the choice of observer set. For example, if the observer set is chosen to contain the union of the two sets we have used in this paper (i.e., for Waves 1 and 2, respectively), then the results will be unchanged. On the other hand, to the extent that a common observer set contains only part or none of the two wave-specific sets we used, the results will change, and can be expected to degrade.

Finally, the framework and the results presented herein allow preliminary delineation of a road-map to set up a surveillance network based on the proposed method in a country. The first step should consist of retrieving data on the spatial distribution of population. Available global sources are, e.g., WorldPop (www.worldpop.org), LandScan (`landscan.ornl.gov`) or the Global Human Settlement Layer (`ghsl.jrc.ec.europa.eu`). Then, possible census data on WASH (Water, Sanitation and Hygiene) conditions, e.g. access to tap water or toilet facilities, should be sought in order to possibly characterize the spatial heterogeneity of the Basic Reproductive number $R_0$. Once such information is collected, the spatially-explicit stochastic epidemiological model can be set up. If data on past outbreaks are available, critical epidemiological parameters can be estimated using such information. Otherwise, reference literature values for such parameters can be assumed. As previously described, simulations of the epidemiological model are used to calibrate the parameters of the probability distributions of first-arrival times. An analysis like the one reported in section *Synthetic experiments* is also recommended to select the best strategy to allocate the observer nodes. Once the number and the location of the observer nodes are decided, an epidemiological surveillance system instructed to routinely perform lab testing for each suspect case of the selected disease is to be established. If an outbreak occurs, data on first arrival times at the selected nodes should readily allow the inference of the possible region of the source of the outbreak, thus enabling fast and effective interventions.

## Methods

### Gaussian source estimation with prior information

Following [12], we cast the source detection problem as identifying the relevant mixture component in a multivariate Gaussian mixture model. However, from the Bayesian perspective we adopt here, whereas the authors in [12] use a uniform prior over sources in their formulation, here we incorporate substantially more structured prior information. This structure arises both through the use of potentially nonuniform priors over sources (i.e., informed by local values of $R_0$) and through calibration of the multivariate Gaussian parameters using a human mobility network and a stochastic epidemiological model.

Let $\pi_s$ be the prior probability of node $s \in V$ being the source and let $\boldsymbol{t}$ be the $K$-dimensional vector of observed first-arrival times. Conditional on $s$ being the true source, $\boldsymbol{t}$ is assumed to follow a multivariate Gaussian distribution, with mean vector $\boldsymbol{\mu}_s$ and covariance matrix $\boldsymbol{\Lambda}_s$. Denote the corresponding density function by $\phi(\boldsymbol{t}; \boldsymbol{\mu}_s, \boldsymbol{\Lambda}_s)$. Then $\boldsymbol{t}$ has density

$$\sum_{j=1}^{N} \pi_j \phi(\boldsymbol{t}; \boldsymbol{\mu}_j, \boldsymbol{\Lambda}_j) \ . \tag{3}$$

A point estimate $\hat{s}$ of the true source, say $s^*$, can be obtained by maximizing the posterior probability computed by Bayes theorem, i.e.

$$\hat{s} = \arg \max_{s \in V} P(S = s | \boldsymbol{t}) = \arg \max_{s \in V} \frac{\pi_s \phi(\boldsymbol{t}; \boldsymbol{\mu}_s, \boldsymbol{\Lambda}_s)}{\sum_{j=1}^{N} \pi_j \phi(\boldsymbol{t}; \boldsymbol{\mu}_j, \boldsymbol{\Lambda}_j)} \ .$$

The formula above can be written as:

$$\hat{s} \quad = \quad \arg \max_{s \in \mathcal{V}} \left\{ -\frac{1}{2} (\boldsymbol{t} - \boldsymbol{\mu}_s)^{\top} \boldsymbol{\Lambda}_s^{-1} (\boldsymbol{t} - \boldsymbol{\mu}_s) + \log \pi_s \right\} \ . \tag{4}$$

Hence, this approach is equivalent to standard linear discriminant analysis for $K$-dimensional

classification, with pre-defined class weights [32]. The Gaussian source estimator by [12] is a special case, assuming a uniform prior, i.e., $\pi_1 = \cdots = \pi_K = 1/K$.

A set estimate $\hat{C}$ for $s^*$ may be obtained in the form of a highest posterior density (HPD) region, by applying the largest threshold $\tau_\alpha$ corresponding to choice of a pre-specified $\alpha$ so that

$$\hat{C} = \{s : P(s|\boldsymbol{T} = \boldsymbol{t}) > \tau_\alpha\}, \text{ so that } \sum_{s \in C} P(s|\boldsymbol{T} = \boldsymbol{t}) \geq 1 - \alpha. \tag{5}$$

The HPD region fulfills the condition that $P(s|\boldsymbol{T} = \boldsymbol{t}) > P(\tilde{s}|\boldsymbol{T} = \boldsymbol{t})$ for all $s \in \hat{C}$ and $\tilde{s} \notin \hat{C}$, and consequently minimizes the volume of the area covered, among all sets with at least $1 - \alpha$ posterior mass [33]. Note that this definition does not consider distance with respect to the network connectivity. Furthermore the HPD region does not necessarily need to be a connected subgraph of the network.

## Parameter calibration using a human mobility network and the stochastic epidemiological model

In order to produce the point and/or set estimates $\hat{s}$ and $\hat{C}$, values must be available for the mean and covariance parameters $\boldsymbol{\mu}_s$ and $\boldsymbol{\Lambda}_s$ of each Gaussian component. There are deterministic estimates available for these parameters, which can be derived easily from network topology information only, using shortest path lengths between potential source candidates and sensors [12]. But $\boldsymbol{\mu}_s$ and $\boldsymbol{\Lambda}_s$—representing first and second order information on the behavior of the first arrival times $\mathbf{t}$—are reflective of what in the current setting is typically a highly complex stochastic phenomenon. Accordingly, we instead calibrate these values in our model using a stochastic epidemiological model that integrates human mobility network information.

A stochastic, spatially-explicit epidemiological model for the transmission of cholera, a prototypical waterborne disease, has been introduced in [22]. This model considers a set of human communities interconnected by a mobility network and describes the temporal evolution of the integer number of susceptible ($\mathcal{S}_i$), infected ($\mathcal{I}_i$), and recovered ($\mathcal{R}_i$) individuals hosted in the nodes $i$ of the network. Additionally, the model incorporates the evolution of the environmental concentration of bacteria ($\mathcal{B}_i$). Events that involve human individuals (i.e., births and deaths, as well as changes in epidemiological status) are treated as stochastic events, each occurring at a rate that depends on the state of the system. The possible events and their corresponding rates are shown in Table 1.

Table 1 shows transitions and rates of occurrence for all possible events indexed by a given node $i$. The generic event $k$ occurs in node $i$ at rate $v_i^k$. Each node has a population that is

Table 1. Transitions and rates of occurrence of all possible events in a node *i*. Based on [22, Table 1].

| Event | Transition | Rate |
|---|---|---|
| Birth | $(\mathcal{S}_i, \mathcal{I}_i, \mathcal{R}_i) \rightarrow (\mathcal{S}_i + 1, \mathcal{I}_i, \mathcal{R}_i)$ | $v_i^1 = \mu H_i$ |
| Death of a susceptible | $(\mathcal{S}_i, \mathcal{I}_i, \mathcal{R}_i) \rightarrow (\mathcal{S}_i - 1, \mathcal{I}_i, \mathcal{R}_i)$ | $v_i^2 = \mu \mathcal{S}_i$ |
| Symptomatic infection | $(\mathcal{S}_i, \mathcal{I}_i, \mathcal{R}_i) \rightarrow (\mathcal{S}_i - 1, \mathcal{I}_i + 1, \mathcal{R}_i)$ | $v_i^3 = \sigma \mathcal{F}_i \mathcal{S}_i$ |
| Death of an infected | $(\mathcal{S}_i, \mathcal{I}_i, \mathcal{R}_i) \rightarrow (\mathcal{S}_i, \mathcal{I}_i - 1, \mathcal{R}_i)$ | $v_i^4 = \mu \mathcal{I}_i$ |
| Cholera-induced death | $(\mathcal{S}_i, \mathcal{I}_i, \mathcal{R}_i) \rightarrow (\mathcal{S}_i, \mathcal{I}_i - 1, \mathcal{R}_i)$ | $v_i^5 = \alpha \mathcal{I}_i$ |
| Recovery of an infected | $(\mathcal{S}_i, \mathcal{I}_i, \mathcal{R}_i) \rightarrow (\mathcal{S}_i, \mathcal{I}_i - 1, \mathcal{R}_i + 1)$ | $v_i^6 = \gamma \mathcal{I}_i$ |
| Asymptomatic infection | $(\mathcal{S}_i, \mathcal{I}_i, \mathcal{R}_i) \rightarrow (\mathcal{S}_i - 1, \mathcal{I}_i, \mathcal{R}_i + 1)$ | $v_i^7 = (1 - \sigma)\mathcal{F}_i \mathcal{S}_i$ |
| Death of a recovered | $(\mathcal{S}_i, \mathcal{I}_i, \mathcal{R}_i) \rightarrow (\mathcal{S}_i, \mathcal{I}_i, \mathcal{R}_i - 1)$ | $v_i^8 = \mu \mathcal{R}_i$ |
| Immunity loss | $(\mathcal{S}_i, \mathcal{I}_i, \mathcal{R}_i) \rightarrow (\mathcal{S}_i + 1, \mathcal{I}_i, \mathcal{R}_i - 1)$ | $v_i^9 = \rho \mathcal{R}_i$ |

assumed to be at demographic equilibrium. We use $\mu$ to represent the human mortality rate, and $\mu H_i$, a constant recruitment rate. The force of infection is defined as

$$\mathcal{F}_i = \beta_i \left[ (1-m)\frac{\mathcal{B}_i}{K+\mathcal{B}_i} + m\sum_{j=1}^{n}Q_{ij}\frac{\mathcal{B}_j}{K+\mathcal{B}_j} \right],$$

and captures the rate at which susceptible individuals become infected due to contact with contaminated water. The parameter $\beta_i$ represents the exposure rate. The fraction $\mathcal{B}_i/(K+\mathcal{B}_i)$ is the probability of becoming infected due to the exposure to a concentration $\mathcal{B}_i$ of *V. cholerae*, $K$ being the half-saturation constant [34]. Because of human mobility, an individual residing at node $i$—if susceptible—can be exposed to pathogens in the destination community $j$. This is modeled in the following way: the force of infection in a given node is assumed to depend on the local concentration $\mathcal{B}_i$ for a fraction $(1-m)$ of the susceptible hosts, and for the remaining fraction $m$, on the concentration $\mathcal{B}_j$ of the surrounding communities. The parameter $m$ thus represents the probability, at a community-level, that individuals travel outside their node (assumed here to be node-independent). The concentrations $\mathcal{B}_j$ are weighted by values $Q_{ij}$, representing the probability an individual living in node $i$ reaches the destination $j$. Matrix $Q$ thus epitomizes information about human mobility. Formally, human mobility patterns are defined according to a gravity model in this approach. $Q_{ij}$ is defined as:

$$Q_{ij} = \frac{H_j e^{-d_{ij}/D}}{\sum_{k\neq i}^{n} H_k e^{-d_{ik}/D}}, \tag{8}$$

where the population size serves as an attractive force, and the distance $d_{ij}$ between two communities (represented using an exponential kernel, with shape-factor $D$), as a deterrent force.

Concentration $\mathcal{B}_i(t)$ is modeled as a stochastic variable in continuous time, based on the expectation of a large number of bacteria. Its evolution is described by:

$$\frac{d\mathcal{B}_i}{dt} = -\mu_B \mathcal{B}_i + \frac{p_i}{W_i}\mathcal{I}_i$$

where $\mu_B$ is the mortality rate of the bacteria in the environment, $p_i$ is the rate at which bacteria produced by one infected person reach and contaminate water in the local reservoir of volume $W_i$, and $\mathcal{I}_i$ is the number of infected.

Assuming a single node $s$ as the source of the epidemic, the stochastic model just described allows for the generation of multiple Monte Carlo realizations of the outbreak. From these realizations we may obtain estimates of the mean and covariance parameters $\boldsymbol{\mu}_s$ and $\boldsymbol{\Lambda}_s$ for the first-arrival times $\mathbf{t}$ at the observers. (Methods of numerical integration or similar might be used here instead). This procedure is repeated assuming each node in turn as a potential source. To estimate $\boldsymbol{\mu}_s$ accurately we rely on large-sample properties of simple averaging. However, our estimation of $\boldsymbol{\Lambda}_s$ was found to benefit from the use of shrinkage methods. We adopted the approach of [35], which assumes zero covariance among off-diagonal elements (supported by our data, most likely due to the sparse and distributed nature of our observer nodes), but heterogeneous variances, which are estimated using a distribution-free shrinkage towards the median.

Additional implementation details may be found in S1 File. In particular, information on how we set the various rate parameters in our stochastic model may be found therein, with corresponding pointers to the supporting literature.

## Supporting information

**S1 Fig. The 2000-2002 cholera outbreak, the first wave.** Node 431: uMhlathuze Local Municipality, South Africa, 143km north of Durban.
(TIF)

**S2 Fig. The 2000-2002 cholera outbreak, the second wave.** Node 450: South west of town Ezakheni A, not far (17km) from Ladysmith to the south east.
(TIF)

**S3 Fig. Sensitivity analysis.** Shows that our results (i.e., the top-ten ranked nodes) remain robust when the choice of threshold is decreased to 0.09% and even 0.05%, but deteriorates at 0.01%.
(TIF)

**S4 Fig. The performance of the proposed method.** (A) empirical coverage probability of the 95% credibility region and (B) the size of the 95% credibility region. Different simulation setting are shown: 9 or 18 observers, random and high-degree observer placement, and incorporation of informative $R_0$ prior and non-informative uniform prior knowledge.
(TIF)

**S5 Fig. The performance of the proposed method.** The probability that the posterior of the true Source is ranked in the Top 10.
(TIF)

**S6 Fig. The performance of the proposed method.** Distance Between the True Source and the Estimation.
(TIF)

**S7 Fig. Comparison between the proposed method and Pinto's method.**
(TIF)

**S8 Fig. Visualizations illustrate that the quality of the Gaussian approximation is quite reasonable, under model assumptions.** To illustrate, we used the 1st-ranked inferred source in the second wave as a source and simulated outcomes according to our model. The marginal distributions for arrival times at the various 2nd wave observers are shown above. We can see that a normal approximation agrees well with the histograms of arrival times.
(TIF)

**S9 Fig. Bivariate plots illustrate the quality of the Gaussian approximation for distributions across pairs of observer nodes.** The bivariate scatterplots agree with the contour of normal distributions with same mean and covariance structure.
(TIF)

**S1 File. Additional details.** [36, 37].
(PDF)

## Acknowledgments

Special thanks go to the KwaZuluNatal Department of Health for providing the data set that made this work possible.

## Author Contributions

**Conceptualization:** Eric D. Kolaczyk.

**Data curation:** Enrico Bertuzzo.

**Formal analysis:** Jun Li, Juliane Manitz.

**Funding acquisition:** Eric D. Kolaczyk.

**Methodology:** Jun Li, Juliane Manitz, Eric D. Kolaczyk.

**Project administration:** Eric D. Kolaczyk.

**Software:** Jun Li, Juliane Manitz.

**Supervision:** Eric D. Kolaczyk.

**Writing – original draft:** Jun Li, Juliane Manitz, Enrico Bertuzzo, Eric D. Kolaczyk.

**Writing – review & editing:** Jun Li, Juliane Manitz, Enrico Bertuzzo, Eric D. Kolaczyk.

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
