## [Decision Letter · Decision Letter 0]

25 Oct 2019

Dear Dr Kolaczyk,

Thank you very much for submitting your manuscript, 'Sensor-based localization of epidemic sources on human mobility networks', to PLOS Computational Biology. As with all papers submitted to the journal, yours was fully evaluated by the PLOS Computational Biology editorial team, and in this case, by independent peer reviewers. The reviewers appreciated the attention to an important topic but identified some aspects of the manuscript that should be improved.  

We would therefore like to ask you to modify the manuscript according to the review recommendations before we can consider your manuscript for acceptance. Your revisions should address the specific points made by each reviewer and we encourage you to respond to particular issues. Please note while forming your response, if your article is accepted, you may have the opportunity to make the peer review history publicly available. The record will include editor decision letters (with reviews) and your responses to reviewer comments. If eligible, we will contact you to opt in or out.raised.

- Supporting Information uploaded as separate files, titled 'Dataset', 'Figure', 'Table', 'Text', 'Protocol', 'Audio', or 'Video'.

We hope to receive your revised manuscript within the next 30 days. If you anticipate any delay in its return, we ask that you let us know the expected resubmission date by email at ploscompbiol@plos.org.

Sincerely,

James Lloyd-Smith

Associate Editor

PLOS Computational Biology

Thomas Lengauer

Methods Editor

PLOS Computational Biology

[LINK]

Reviewer's Responses to Questions

**Comments to the Authors:**

Reviewer #1: Please see the comments in attachment.

Reviewer #2: Source location is an important problem, especially for human disease epidemics, because one is not only to confronted with limited information from a sparse set of observers, but also with very partial and noisy knowledge on the percentage of infected individuals in a community, on the time of their first infection, and above all on the contact network, which is even time-varying because of human mobility. The paper presents a well written, very interesting case study of source location for two waves of a cholera outbreak in the KwaZuku-Natal province of South Africa.

Depending on the information available at the nodes (representing here communities) of the network, most source location methods rely on having either only information about the state (susceptible, infected and possibly recovered) of all or most of the nodes at some point in time (snapshot methods), or on having also information about the infection time but only at a small set of sensors (sensor, time-based methods), as in this paper. The paper combines the sensor, time-based source location method from [12] with priors on the location of the source and on the contact graph, which results from models developed by some of the authors in [21,23]. The source location methods are not so much different, what strongly differs are the priors. The estimation (4) of the source is the same as eq. (2) in [12], but the latter uses only default priors, in absence of better information: the source location prior is uniform among the nodes, and the parameters (mean delay \\mu, delay covariance \\Lambda) of the Gaussian distribution of the delays are set using only paths lengths. The current paper uses well crafted model- and data-driven priors (which apparently might even depend on the epidemic wave? see question below). The source location prior is obtained from a local epidemiological reproduction number R_0 for each community, and the parameters (\\mu, \\Lambda) are obtained by calibrating of a simulated SIR model whose rates are obtained using community populations and a gravity mobility model between them. The good quality of these data-driven priors seems to be main factor that significantly improves the location of the source, more than the constraint of a breadth-first search propagation tree. To have a comparison with a different method, the authors would therefore have picked another different sensor-based method listed e.g. in [5].

The paper shows that a sensor, time-based source location method can have good performance but needs to have good model- and data-driven priors to be effective, which is a very nice contribution to the state of the art. The model used to derive these priors is well explained in terms of human mobility and epidemiology.

Some questions would however need to be clarified:

- What is the network topology resulting from the calibration of the parameters (\\mu, \\Lamba)? Is it different than a complete graph with different values of \\mu along the edges of the graph, or did you use a threshold on \\mu to cut some edges?

- What is the threshold used to declare a community infected and to set the entries of the first-arrival times vector t? Is it 0.1% of the population as for the source node? How sensitive are the results to this threshold?

- What is the ground-truth for the two waves? In Fig. 6 (respectively, Fig. 7), Node 431 (respectively, 450) is highlighted. Are these nodes the actual sources? Node 431 appears far from the source candidates, contrary to Node 450.

- Are the parameters of the model depending on the epidemic wave? The choice of the observers is different between both waves in Figs 2 and 3 (also, is the prior on the source uniform in Fig 3, as it is in Fig 2?). With the default priors, the parameters and observers would be independent of the epidemic wave, and hence can be set once for all, before the outbreak of the epidemics. What would the results be if the observers (and parameters of the model) were the same for both waves?

**Have all data underlying the figures and results presented in the manuscript been provided?**

Reviewer #1: Yes

Reviewer #2: No: See authors' answer to this question: the authors received the data from the Kwa-Zulu Natal Dept of Health in South Africa, with a restriction that they could not redistribute the data but could point people to them.

PLOS authors have the option to publish the peer review history of their article (what does this mean?). If published, this will include your full peer review and any attached files.

Reviewer #1: No

Reviewer #2: Yes: Patrick Thiran

---

## [Decision Letter · Decision Letter 1]

15 Sep 2020

Dear Prof. Kolaczyk,

Thank you very much for submitting your manuscript "Sensor-based localization of epidemic sources on human mobility networks" for consideration at PLOS Computational Biology. As with all papers reviewed by the journal, your manuscript was reviewed by members of the editorial board and by several independent reviewers. The reviewers appreciated the attention to an important topic. Based on the reviews, we are likely to accept this manuscript for publication, providing that you modify the manuscript according to the review recommendations.

In particular, reviewer 2 makes a number of specific recommendations for further changes needed to the manuscript's text (and maybe an addition to the supplement).  If you agree with these changes, they seem like they would be straightforward to implement.  If you don't agree, please provide a full rationale and expect that the manuscript will be seen by this reviewer again. 

Reviewer 1 sets the one condition that the software package must be made publicly available, which is a position shared by the journal of course.

Sincerely,

James Lloyd-Smith

Associate Editor

PLOS Computational Biology

Thomas Lengauer

Methods Editor

PLOS Computational Biology

[LINK]

Reviewer's Responses to Questions

**Comments to the Authors:**

Reviewer #1: Thank you for your very thorough and very clear point-by-point response to my comments, which greatly facilitated my ability to review.

The article provides an important next step to the field of source detection -- improving upon the first wave of source detection methods, which were mathematically parsimonious and un-specific to context, and bringing in improvements to enable customization to specific epidemiological contexts through more sophisticated methodological enhancements to account for network spreading structure, prior information, and accuracy evaluation framework.

I have no further comments besides the importance of publishing the NetOrigin source code package.

Reviewer #2: The review comments for the first version of the paper have been only partially addressed. I keep considering the paper as an interesting piece of work, but the following points still need to be addressed.

1) As mentioned before, the methods developed in this paper and in [12] are not really different, what differs are the priors: the estimation (4) of the source is the same as eq. (2) in [12], but the latter uses only default priors. Therefore, the section "Gaussian source estimation with prior information" should mention more precisely that the authors' method extends the source detection problem formulated as the identification of the relevant mixture component in a multivariate Gaussian mixture model from a uniform prior in [12] to general priors, instead of stating that they recast the source detection problem as identifying the relevant mixture component in a multivariate Gaussian mixture model.

2) It is the good quality of the data-driven priors obtained by the authors, which significantly improves the location of the source, not so much the breadth-first search propagation tree as in [12]. Yet, the discussion section maintains that the improvement may also be due to the requirement that the network takes the form of a tree. But this is a requirement made in [12] to have a mathematically proven guarantee that the estimator (4) (i.e., (2) in [12]) is optimal, because optimality does not necessarily hold for meshed networks. The method in the current paper is therefore also optimal for a known propagation tree, but the paper does not give a proof that the estimator (4) is optimal for general networks either. This part of the discussion should therefore be rephrased.

3) Thanks for the clarification that the values of the parameters (\\mu, \\Lambda) are source-node-indexed and not edge attribute, but it is still not clear what the contact graph is. The authors do not state it in their answer to my question, but in the revised version, I read that "more formally, the network visualization suggests small-world behavior, which is confirmed through computational methods (e.g., [31, Ch5.5.2], analysis not shown). How confident can we be in the authors' suggestion: Is the contact network a small-world network? if so, this analysis would more than welcome in the supplemental material.

4) Thanks for the results on the sensitivity of the results to the threshold of 0.1% of the population. It would be good to mention in the paragraph added just before the results on synthetic experiments (and not only in the authors' response) that thresholds larger than 0.1%, even 0.2% makes the inference procedure fail since not all observers are infected.

5) Thanks for clarifying that there is no ground-truth for the two waves, this explanation would also benefit to be stated at least in the supplemental material.

6) Thanks for the discussion on the dependence of the parameters of the model on the epidemic wave. Some of the discussion should be included in the paper and in the supplemental material, and in particular, that the parameters and observers would be independent of the epidemic wave with default uniform priors (as in [12]), and hence can be set once for all, before the outbreak of the epidemics. Now, with some history on multiple epidemics (or other side information), it is very likely to have much better priors than uniform, in which case they should be used as in this paper.

**Have all data underlying the figures and results presented in the manuscript been provided?**

Reviewer #1: Yes

Reviewer #2: Yes

PLOS authors have the option to publish the peer review history of their article (what does this mean?). If published, this will include your full peer review and any attached files.

Reviewer #1: **Yes: **Abigail L. Horn

Reviewer #2: No
---

## [Editor Report · Decision Letter 2]

17 Nov 2020

Dear Prof. Kolaczyk,

We are pleased to inform you that your manuscript 'Sensor-based localization of epidemic sources on human mobility networks' has been provisionally accepted for publication in PLOS Computational Biology.

Best regards,

James Lloyd-Smith

Associate Editor

PLOS Computational Biology

Thomas Lengauer

Methods Editor

PLOS Computational Biology

---

## [Editor Report · Acceptance letter]

22 Jan 2021

PCOMPBIOL-D-19-01206R2 

Sensor-based localization of epidemic sources on human mobility networks

Dear Dr Kolaczyk,

I am pleased to inform you that your manuscript has been formally accepted for publication in PLOS Computational Biology. Your manuscript is now with our production department and you will be notified of the publication date in due course.

With kind regards,

Jutka Oroszlan
